# Neuromodulation of Neural Oscillations in Health and Disease

**DOI:** 10.3390/biology12030371

**Published:** 2023-02-26

**Authors:** Evan Weiss, Michael Kann, Qi Wang

**Affiliations:** Department of Biomedical Engineering, Columbia University, ET 351, 500 W. 120th Street, New York, NY 10027, USA

**Keywords:** neuromodulation, EEG, noradrenergic system, cholinergic system, dopaminergic system, pupil-linked arousal, neural oscillations, cross-frequency coupling, vagus nerve stimulation, neurological disorders

## Abstract

**Simple Summary:**

Over the past few decades, advances in electroencephalography (EEG) recordings and brain stimulation has permitted an unprecedented view of how specific brain structures communicate as well as organize complex cognitive functions. Specifically, neurotransmitters (including norepinephrine, acetylcholine, and dopamine) have all been shown to have an impact on neural oscillations throughout the brain, linking them to changes in cognitive functions such as memory, attention, and executive function. While these interactions are still widely unexplored, their appearance in neurological disorders through cross-frequency coupling (CFC) brings light to the vital role they play in orchestrating healthy brain function. This brief review serves to highlight the important role each neuromodulatory system plays in changing widespread neural networks, emphasizing their involvement in health and disease to help inform more translational brain stimulation technologies.

**Abstract:**

Using EEG and local field potentials (LFPs) as an index of large-scale neural activities, research has been able to associate neural oscillations in different frequency bands with markers of cognitive functions, goal-directed behavior, and various neurological disorders. While this gives us a glimpse into how neurons communicate throughout the brain, the causality of these synchronized network activities remains poorly understood. Moreover, the effect of the major neuromodulatory systems (e.g., noradrenergic, cholinergic, and dopaminergic) on brain oscillations has drawn much attention. More recent studies have suggested that cross-frequency coupling (CFC) is heavily responsible for mediating network-wide communication across subcortical and cortical brain structures, implicating the importance of neurotransmitters in shaping coordinated actions. By bringing to light the role each neuromodulatory system plays in regulating brain-wide neural oscillations, we hope to paint a clearer picture of the pivotal role neural oscillations play in a variety of cognitive functions and neurological disorders, and how neuromodulation techniques can be optimized as a means of controlling neural network dynamics. The aim of this review is to showcase the important role that neuromodulatory systems play in large-scale neural network dynamics, informing future studies to pay close attention to their involvement in specific features of neural oscillations and associated behaviors.

## 1. Introduction

Neural oscillations are thought to be an essential driver of interaction, communication, and information transmission throughout the brain [1,2,3]. Evolution has maximized the role these oscillations play in regulating and controlling neuronal functions, driving the synchronization of widespread neural networks in the brain. The EEG provides the most popular non-invasive methods to record neural oscillations, summating the local field potentials of thousands of neurons in cortical structures [4,5]. Not only does this tool characterize the “global” brain state as a time series of voltage potentials, but also allows researchers to analyze these oscillatory waveforms through frequency domain analysis. Studies have suggested that distinct EEG frequency bands (Delta, Theta, Alpha, Beta, Gamma) are generated from unique neural populations across a variety of brain regions [6]. This characterization shows the ability of different brain structures to generate specific neural oscillatory patterns, permitting synchronization and frequency coupling [6]. Assessing the effect of oscillatory changes both globally and locally throughout the brain can uncover the important and/or causal role of various neuromodulatory processes [7]. Cross-frequency coupling (CFC), resulting from coupling between various neural circuits and/or different types of neurons through chemical or electrical synapses, has recently become a more prominent topic [8,9]. Components of CFC such as phase–phase and phase–amplitude coupling have been shown to have a large influence on cognitive processes including attention, learning, and short- and long-term memory [10,11]. Additionally, the phase–amplitude synchronization of these high and low frequency bands plays a prominent role in facilitating neural communication and neural plasticity [12,13,14,15]. Neurological diseases and conditions can often be associated with an abnormal oscillatory desynchronization or type of CFC, unveiling the importance that synchronized neural networks have in carrying out normal brain function [16]. While largely misunderstood, this network-wide communication seems to be an instrumental part of the coordination and regulation of cognitive abilities. More recently, neural stimulation techniques such as vagus nerve stimulation (VNS) and deep brain stimulation (DBS) have been incorporating types of CFC analysis to better understand the effects of phase-coupled neuromodulation [17,18,19,20]. By leveraging the causal effects of neuromodulation on cognitive functions, these tools are focusing on real-time oscillation analysis to optimize the effectiveness of stimulation across brain regions [21].

The neuromodulatory systems, including the noradrenergic, cholinergic, and dopaminergic systems, play a pivotal role in the regulation and synchronization of neural oscillations. These systems provide direct axonal projections to most structures of the brain, regulating various brain functions through the release of neurotransmitters [22,23,24,25] (Figure 1a). Specifically, norepinephrine, acetylcholine, and dopamine are all implicated in the formation of complex decision making and executive functions [26]. Through their activation and inhibition, each neuromodulatory system has been seen to change the oscillatory behavior of widespread neural networks, implicating changes in cortical structures as well as in different frequency bands [27]. Recently, more work has been conducted that focuses on how cross-frequency coupling can be affected by neuromodulation, with phasic or tonic neurotransmitter release causing synchronization or desynchronization in EEG waveform features such as power, amplitude, phase and frequency [28]. These experiments are often using optogenetic manipulation in conjunction with LFP recordings, providing insights into how particular neuromodulatory centers can have a profound effect on neural oscillations, even in indirectly coupled brain regions. Another non-invasive biometric measure, pupil size, has also been implicated in having a key role in indexing neuromodulation, potentially serving as a new indirect modality in understanding the widespread effect of different arousal states on neural oscillations and behavior [29,30,31,32,33,34].

This review will focus on how these three neuromodulatory systems can individually modulate neural oscillations, looking specifically at how their activation can change large-scale neural network synchrony in cognitive functions and neurological disorders. This phenomenon has yet to be fully understood, and looking at how each system’s activation or inhibition affects large scale oscillatory patterns may help uncover the origin and causality of complex behaviors and neurological disorders. These insights will hopefully inform a new direction of research that looks to further investigate how neuromodulation could improve and/or shape brain functions through changing large-scale neural oscillations.

## 2. Neural Oscillations in Cognition and Neurological Disorders

### 2.1. Frequency Spectral Analysis

Over the past few decades, an abundance of research has been conducted exploring the role different frequency bands play in representing the neural signals behind cognition, behavior, and neurological disease. These have been characterized by abstracting the frequency spectrum into five unique frequency bands: Delta (1–3 Hz), Theta (4–7 Hz), Alpha (8–12 Hz), Beta (13–30 Hz), and Gamma (30–100 Hz) [35]. While an abundance of studies has implicated each band in a variety of diseases and cognitive functions, generally, slow and high oscillations can be discriminated by specific behavioral outputs. It is important to acknowledge that these oscillations are often shaped by the summations of thousands of neurons in a particular brain region, making these associations highly dependent upon the type of recording device (EEG, ECOG, invasive LFPs) and their signal-to-noise ratio. Even with this in mind, looking at how each frequency band is implicated throughout complex cognitive processes can make it easier to understand how neuromodulation plays a vital role in shaping these neural oscillations.

#### 2.1.1. Healthy Functions

Throughout normal cognition, each oscillatory frequency band has been linked to a specific behavioral correlate or executive function. Generally speaking, delta waves are associated with a wide variety of cognitive functions. Studies have shown that the delta oscillatory phase is correlated with the reaction time of behavior, playing a role in the synchronization of neural populations across multiple brain regions [36]. Additionally, these oscillations are often associated with a resting state, with the slow oscillations (less than 1 Hz) having the ability to trigger thalamically-generated spindles [37]. Theta oscillations are thought to play a vital role in almost every cognitive function and brain region. Studies have uncovered their critical role in learning, memory, and synaptic plasticity. It is also a foundational band for cross-frequency coupling, synchronizing with the gamma band in a phenomenon known as theta–gamma coupling that is heavily studied for its role in working memory [38,39,40,41]. The alpha band is also one of the most studied frequency bands, specifically for its clear role in attentional demands and visual perception as well as its easily triggered appearance in stimuli-associated tasks [42,43,44,45,46,47]. The beta band has often been associated with motor-induced events [48]. The synchronization of beta frequencies has been an indicator for normal motor system functioning, and it has been shown to play a role in working memory [49,50]. Finally, the gamma band, like the theta band, plays a vital role in memory, as seen in theta–gamma coupled oscillations throughout the hippocampus and cortex [41]. Gamma also has appearances in attention, consciousness, and wakefulness experiments [51,52,53].

#### 2.1.2. Neurological Diseases

While each frequency band is instrumental to maintain healthy brain function, neurological diseases seem to alter specific oscillatory properties, oftentimes leading to cognitive dysfunction. To begin, abnormal delta oscillations have often been linked to neurological conditions associated with sleep. Specifically, non-rapid eye movement sleep in both Schizophrenia and Alzheimer’s disease has been associated with a depressed delta power when compared to healthy patients [54,55] Clinically, the theta band has been linked to the maintenance of the healthy human brain. Studies have explored the loss of long-range temporal correlations in theta oscillations in patients with major depressive disorder as well as increased theta event-related synchronization in children diagnosed with ADHD [56,57]. A slowing in spontaneous alpha oscillations has been consistently linked to Alzheimer’s disease. Additionally, likely due to its prominent role in attention, resting alpha power has been shown to be reduced in adults with ADHD [58,59]. Higher frequency oscillations have been largely linked to motor-related conditions such as Parkinson’s tremors. The beta band has been reported to be desynchronized in Parkinson’s patients [48]. The modulation of beta oscillations through deep brain stimulation has enabled new targeted therapeutic treatments, oftentimes reversing these motor symptoms [60]. Gamma oscillations have also appeared in Parkinson’s disease and during dyskinesia [61]. Likewise, aberrant gamma oscillations have appeared in mouse models of Alzheimer’s disease and Fragile X syndrome [62].

### 2.2. Cross-Frequency Coupling Analysis

While each frequency band seems to dissociate into a wide variety of behaviors or cognitive functions, the synchrony between each band through cross-frequency coupling (CFC) analysis has brought new light to the complexity of neural networks [63]. Recent studies which uncover the coupling between distinct features of neural oscillations such as phase, amplitude, and power between frequency bands have shown its critical importance to network dynamics, learning, and complex behaviors [64,65,66,67]. CFC is thought to be integral to the temporal and spatial activation of specific cortical circuits, with one study showing how the magnitude of gamma oscillations are modulated by slower wave rhythms [68]. By pairing each frequency interaction with another (i.e., power–power, phase–phase, phase–frequency, phase–power, and phase–amplitude), each coupled oscillatory feature provides a unique effect on the synchronization of widespread neural networks and changes in cognitive abilities [10,69,70,71,72,73]. For example, observed phenomena such as theta–gamma coupling, known for its role in working memory, is only one of many oscillatory patterns which has linked the important role of band coupling in neural communication, synaptic plasticity, and executive functions [8,10,40,74,75,76,77] (Figure 1b).

#### 2.2.1. Healthy Functions

Previous studies show more instances of CFC playing a role in cognitive functions across memory, learning, attention, and decision making. Specifically, alpha–beta phase–amplitude coupling (PAC) has been seen synchronized in the medial prefrontal cortex during decision making, as well as the phase of delta and theta bands, suggesting the role of PAC in feedback coding [78,79]. In decision making, CFC was seen to be highly correlated with rodent behavioral performance, with CFC strength increasing over time in hippocampal regions [65]. PAC was also seen between alpha and gamma bands in response to visual grating stimuli [80]. Additionally, attention has been seen in a wide variety of CFC phenomena, with purposeful synchronization across brain regions driving the formation of neural ensembles associated with specific tasks. Specifically, spatial attention experiments showcase two unique PAC clusters, with delta–gamma PAC being sensitive to cue direction and theta–alpha and beta–gamma PAC associated with future reaction times [81]. Similarly, delta rhythm synchronization was correlated with the reaction time to anticipatory signals [36]. Alpha and gamma rhythm synchronization has also been seen to contribute to selective attention, with a study showing increased beta and alpha synchronization in rule-directed task behaviors [82]. Delta–theta phase high gamma amplitude coupling, triggered by attentional demands, was seen as a mechanism for sub-second facilitation and coordination in the parietal cortex [83]. Likewise, the implementation of alpha oscillations in visual discrimination highlights the role of CFC in filtering out stimuli distractors [84].

CFC is also thought to be the brain’s method of transferring large amounts of information across distant regions of the brain in an organized fashion, and it has been shown to impact different forms of learning and memory across species. Theta oscillatory synchronization between the prefrontal cortex (PFC) and the amygdala supports communication across interneuron networks and is associated with fear learning [85]. Theta band synchronization of the anterior limbic system was also shown to be associated with long term memory, while alpha band desynchronization of the posterior thalamic system was associated with short-term memory [86]. Additionally, synchronization of the PFC between 3 and 32 Hz aided in short-term memory [87]. The large variety in synchronization patterns found across short- and long-term memory could be a result of the lack of consistency in behavioral tasks and experimental methods used across diverse research groups. Nevertheless, it could also showcase that optimal short- and long-term encoding for unique memory tasks or objects that occurs at very specific CFC patterns [87]. It is important to note that the CFC studies that we just described were characterized throughout healthy brain function and observed during the natural rhythms of wide-spread neural networks. While these oscillatory phenomena have uncovered interesting links to cognitive functions, their causality is still mostly untested. It is possible that CFC is a predictive biomarker of communications between distinct populations of neurons mediating various brain functions and is therefore subject to the influence of neuromodulatory systems.

#### 2.2.2. Neurological Diseases

In playing such a large role in neural physiology, it is no surprise that CFC has also been involved in a variety of neurological diseases. Being most observed in working memory studies, PAC has been associated with diseases including schizophrenia, obsessive–compulsive disorder, Alzheimer’s disease, epilepsy, and Parkinson’s disease. For example, it has been suggested that CFC may play a role in Parkinson’s disease as excessive PAC between beta and gamma bands has been observed in advanced-stage patients with movement deficits [64,88]. Abnormal theta–gamma CFC has been implicated in Alzheimer’s disease, as coupled band synchronization plays a critical role in memory functions [89,90]. Also, theta–phase gamma–amplitude coupling has been seen as a marker of Attention Deficit Hyperactivity disorder (ADHD) in children [91]. Likewise, schizophrenia studies have revealed adverse CFC in healthy patients as they suffer from clear disruptions in theta and gamma oscillations in the temporal lobe and auditory cortex [92,93]. Disease-driven CFC is rooted in the loss of synchronization between frequency bands that are in coordination in a healthy state. These disruptions are often a product of pathological interference, preventing normal brain function due to a lack of synaptic connections or adverse oscillatory rhythms. While CFC’s role in disease is far from understood, future experimental models should make note of CFC changes, as mapping out these oscillatory dynamics could help enable faster diagnosis and/or more targeted brain stimulation treatment.

Cross-frequency coupling has been found in the synchronization across subcortical and cortical regions, orchestrating a bottom-up circuit where deep brain regions such as the hippocampus and neuromodulatory centers such as the locus coeruleus (LC), ventral tegmental area (VTA), and basal forebrain (BF) have direct control of brain-wide neuronal synchrony. However, this phenomenon has yet to be explored in patients, as invasive stimulation and optogenetic/chemogenetic manipulations are challenging to conduct in human studies. Future experiments using animal disease models would be better positioned to look into how specific brain structures modulate CFC to help uncover the causal relationship between cognitive abilities and neural oscillations in health and disease.

## 3. The Neuromodulatory System’s Role in Neural Oscillations

### 3.1. The Noradrenergic System

The noradrenergic system plays a very important role in the modulation and regulation of norepinephrine (NE) throughout the brain. The center of norepinephrine creation, the locus coeruleus, is a complex nucleus with projections to almost every area of the cortex and subcortical areas [94,95,96,97]. Projections directly from the LC paired with directed synchronous and asynchronous firing patterns of LC neurons have been shown to allow for differentiated and targeted norepinephrine signaling throughout the cortex [96,98] (Figure 1a). The LC releases norepinephrine via two patterns: tonic and phasic firing. Phasic activation, which is commonly seen in rapid behavioral adaptation, is often associated with a large-scale reorganization of targeted neural networks [99,100]. Tonic activation, in contrast, seems to have a prominent role in NE release but lacks the desynchronization associated with phasic activation [101]. Differences within and across phasic and tonic stimulation patterns in LC neurons is implicated in various types of behavioral tasks and attentional processing [102]. While studies have not specifically defined the LC’s direct role in widespread neural oscillations, experiments incorporating a type of LC modulation in conjunction with electrophysiology recordings have unveiled the LC’s ability to control selective neural oscillatory patterns all the way down to specific frequency bands in localized brain regions. Phasic activation of the LC has been shown to result in clear desynchronization in the EEG spectrum [103,104]. For instance, phasic microstimulation of the LC of rats increased the ratio of EEG power in high frequencies (10–100 Hz) to low frequencies (1–10 Hz), desynchronizing cortical EEG [105] (Figure 2a,b). An abundance of work has also been performed focusing on the LC’s role in the hippocampus, with the LC-NE system seen driving long-term potentiation (LTP), increasing theta power, and with gamma power being reduced in rats that did not experience LTP [101]. Each type of noradrenergic receptor, α1, α2, and β1, has been implicated in theta oscillations and synchronization in the hippocampal formation [106,107,108,109]. Furthermore, studies have shown that theta activity was able to be modulated by reboxetine, a norepinephrine reuptake inhibitor [110]. In regard to cortex-specific cognitive functions, phasic release of norepinephrine has been seen to change the EEG spectrum across the mPFC, guiding motor planning, decision making, and sensory processing [111,112,113]. The activation and inhibition of subcortical regions has led to alpha wave synchronization which facilitates in directing selective attention [114]. The bidirectional communication between the LC and PFC has been seen to act as a level of control over excitatory input into the LC, marked by a high gamma frequency in the PFC [28]. Optogenetic stimulation of the LC in conjunction with neural spike recording in the mPFC saw increased neuronal firing and persistent spikes, present in high amplitude and slow frequency oscillations (delta and theta waves). These increased depolarization events are often associated with an enhancement of synaptic plasticity and memory consolidation [115]. Taken together, these results suggested that the coupling of phase, amplitude, and power between different frequency bands may be directly influenced by NE release from the LC, with specific stimulation parameters being essential for changes in the synchronization of these oscillatory features. Some recent evidence showed that blocking the noradrenergic system through administration of Clonidine, an alpha-2 agonist, changed the phase–amplitude coupling in cortical EEG in mice (Figure 2c). Future studies which look to selectively adjust the frequency and power of stimulation, directing a graded release of NE, could potentially provide a type of control over neural oscillations or help restore CFC relationships that may have been lost to neurological disorders. Although a substantial amount of research questions remain unanswered about the true relationship between the noradrenergic system and neural oscillations, these previous studies open the door to a clear causal relationship of the role NE plays in shaping cognitive abilities through widespread oscillatory dynamics.

### 3.2. The Cholinergic System

The cholinergic system, like the noradrenergic system, also plays a vital role in the control of neural oscillations throughout the brain and cortical structures [116]. Studies have already unveiled its contribution in central nervous system physiology and in various disorders such as dementia, epilepsy, and sleep disorders [117,118,119]. While acetylcholine (ACh) is more widespread and abundant in the brain than norepinephrine, its neuromodulatory centers, including the pedunculopontine nucleus (PPT), laterodorsal tegmental nucleus (LDT), and basal forebrain (BF), have direct projections to widespread brain regions, allowing the cholinergic system to readily influence the oscillations of neural networks [26]. Retrograde tracing has shown that the cortex has an abundance of projections from the basal forebrain, connecting subcortical structures to the frontal, cingulate, and medial parietal cortex [120,121] (Figure 1a). Specifically, the BF has been seen implicated in the direct regulation of attentional functions through multiple thalamic and cortical projections [122]. Similar to the release of norepinephrine from the LC, acetylcholine is also released throughout the brain in tonic and phasic patterns [123,124]. Studies have shown that tonic acetylcholine release in the prefrontal cortex was coordinated with the hippocampus and was maximal during training on a working memory task. Phasic release, in contrast, only occurred during the memory task and was localized to reward delivery areas, independent of the trial outcome [125]. Although tonic, volume-based ACh release is traditionally viewed to be the driver, recent optogenetic studies suggested a link between phasic ACh activation in the cortex and its causal role in behavioral changes [126].

Like noradrenergic modulation, cholinergic modulation has not specifically been studied in conjunction with cross-frequency coupled phenomena. Instead, studies utilizing widespread neural recordings have looked to uncover the direct role of ACh in behavior and erratic firing patterns associated with neurological disorders. Specific low-frequency oscillations including delta, theta, and alpha have been seen to be suppressed by brainstem cholinergic neurons, while high-frequency bands such as beta and gamma are accompanied by an increased release of ACh in the thalamus and cortex [127]. Supporting this notion, direct electrical stimulation of the nucleus basalis, the cholinergic nucleus of the basal forebrain, increased the ratio of power in high frequency (10–100 Hz) bands over low frequency (1–10 Hz) bands of cortical LFPs [128] (Figure 3a,b). ACh release in cholinergic neurons has also been seen to discharge at higher rates during cortical activation rather than during slow-wave cortical activity, with thalamo-cortical and brainstem-cortical cholinergic activity initiating theta rhythms and influencing task-specific cortical desynchronization [129]. Cholinergic system modulation is also highly dependent upon input from noradrenergic LC projections. A particular study explored that when noradrenaline was administered into the basal forebrain in sleep-waking rats, it elicited an increase in fast gamma EEG oscillations and a diminution in slow delta activity, but when neurotensin was administered, theta and gamma activity as well as wakefulness were enhanced [130,131]. Furthermore, cholinergic neurons seem to play a vital role in influencing theta oscillations in the hippocampus. Studies have shown a direct relationship between ACh level and theta oscillations in hippocampal neurons, as well as a direct impact on the amplitude of theta waves. This relationship potentially influences neural computations responsible for memory encoding and retrieval [132,133,134,135]. Theta–gamma coupling also seems heavily dependent on ACh modulation, with detected cues evoking phasic ACh release as well as neural synchrony between frequency bands in the PFC [126,136]. Likewise, varying regions of high and low ACh signaling lead to the emergence of localized gamma–theta coupling, lending support to the theory of cross communication between brain regions during attentional processes. Specifically, studies have shown that stable gamma-modulated firing occurs in regions with high ACh signaling, implicating its causal role in generating localized theta–gamma rhythms [51].

Oftentimes, Ach-triggered neural synchrony has been explored using activation or inhibition studies through optogenetics and pharmacological agents. Specifically, phasic optogenetic stimulation of the basal forebrain has been seen to modulate the cortical topography of auditory steady state responses, with phase-locked stimulation enhancing the power of cortical responses and increasing their synchronization, as well as changing broadband gamma power [137,138,139]. Additionally, stimulation applied to the pedunculopontine tegmentum (PPT) significantly increased Ach release that drove desynchronization in the cortical EEG [140]. The relationship between CFC and cholinergic modulation is still a widely unexplored phenomenon, but future research specifically looking at how phasic stimulation parameters could affect different aspects of CFC could provide new insights into how the cholinergic system drives complex cognitive functions through widespread neural oscillations.

### 3.3. The Dopaminergic System

The dopaminergic system is critically important to the circuitry and control of cognitive functions in the prefrontal cortex [141]. Like the other neuromodulators, dopamine is synthesized in various centers across the brain including the ventral tegmental area (VTA), the hypothalamus, and the substantia nigra (SN) [24] (Figure 1a). Theories have proposed a complex neuronal microcircuit based on the known mechanism of action of dopamine in the PFC, accounting for the diverse role of dopamine in executive functions [142]. The direct projections from the VTA and SN via the mesocortical dopamine pathway are responsible for functions such as working memory, attention, and decision making [143]. Similar to the release of other neuromodulators, tonic and phasic dopamine release have been shown to play different roles in mediating a wide range of brain functions, including learning, motivation, and motor control, possibly through their distinct effects on dopaminergic receptors [144].

The dopaminergic system has been shown to be heavily involved in the shaping of neural oscillations in a variety of cognitive functions and neurological disorders. Studies utilizing electrophysiology recordings have demonstrated the vital role of dopamine activation throughout the brain. Specifically, dopaminergic receptor activation has been linked to weaker alpha and beta oscillations in the PFC, while depletion of dopamine has been linked to increased power in beta oscillations in the cortex and subthalamic nucleus of rats [145,146]. Dopamine also plays a role in working memory, with it facilitating low theta oscillations in the PFC as well as acting as a trigger for latent theta oscillations [147]. Dopaminergic modulation through stimulation and pharmacology has also seen clear effects on cortical state. Dopamine injections in the PFC of anesthetized rats, for example, has been shown to provoke an increase in hippocampal and prefrontal coherence [148]. Stimulation of the VTA has been shown to induce reanimation from anesthesia, with optogenetic stimulation of even a small portion of VTA dopamine neurons being sufficient to induce this transition [149]. Additionally, the emergence of exaggerated beta oscillations in the cortex, a key symptom of Parkinson’s disease, is marked by a disruption in dopamine transmission [150]. The synchronization of beta frequencies facilitates normal motor functioning, and dopamine levels largely impact beta synchronization to guide normal function [49]. However, while beta oscillations in cortical and basal ganglia networks are closely coupled to dopamine tone in humans, phase–amplitude coupling appeared not to be directly regulated by dopamine levels. These findings have key implications for Parkinson’s patients, and future research is necessary to uncover how these findings can be used to benefit patients’ quality of life [151].

The dopaminergic system clearly acts as a controller of large-scale neural oscillations, but its causal relationship with CFC is largely unknown. Looking at dopamine-triggered synchronization studies may help provide context to the role of dopamine in CFC. In Parkinson’s patients, chronic dopaminergic transmission interruption promotes excessive cortical beta synchronization that is seen in high coherence between motor and somatosensory cortical activities [152]. Desynchronization in the cortex has been seen through the stimulation of D1 dopamine receptors in behavioral arousal experiments in rats and rabbits [153]. Looking into more specific phase and amplitude phenomena, dopamine release in the mPFC has led to shifts in phase–amplitude coupling from the theta–gamma band to delta–gamma band, giving insight into how dopamine may regulate function in the mPFC [72]. Interestingly, in this study, laser activation of RuBi-Dopa, a light-sensitive caged compound, released dopamine and in turn increased activation of dopamine receptors, resulting in a modulation of CFC throughout LFP recordings in the mPFC but not LFP power in any frequency band [72] (Figure 3c). Furthermore, theta phase coupling between the hippocampus and the mPFC may also be modulated by the dopamine system and could be an underlying mechanism of cognitive dysfunction in depression [154]. Recently, a study showed how phasic dopamine activation in the PFC had a robust influence on coordinated gamma oscillations, initiating a gamma–theta coupling which lasted for several minutes [155]. Overall, the relationship between cross-frequency coupling and dopamine is still widely unclear. Future research studying how stimulation of dopamine neuromodulatory centers and receptors impact select features of neural oscillations could further help us understand the causality of cognitive functions and neuropsychiatric disorders.

## 4. Pupil-Linked Arousal and Neural Oscillations

While future research needs to be conducted exploring each neuromodulatory system’s direct role in neural oscillations, looking at another non-invasive metric, such as the pupil, could provide insight into the interaction between neuromodulation and cognition. Previous work has showcased the ability for the pupil to be used as a non-invasive readout of the central arousal state (pupil-linked arousal), with possible involvement of the noradrenergic and cholinergic systems [29,30,32,105,156]. In both humans and rodents, the pupil has been found to have a clear role in representing phasic arousal levels in perceptual decision-making tasks [34,157,158,159,160,161]. While studies have not yet explored the direct relationship between pupil and cross-frequency coupling, many have started to use pupil/oculomotor dynamics as an additional readout of brain state. For example, a study showcased how CFC between the phase-locked amplitudes of the gamma band and delta band corresponded to success in a visual discrimination task [162]. However, they failed to find that eye movements have any significant effect on the cross-frequency coupling in the EEG. An additional study explored the pupil dynamics in LFP recordings in the lateral hypothalamus, revealing a clear correlation between pupil dilation events and delta power, opening the door to new questions around pupillary changes and brain state transitions [163]. These studies raised new questions about the role of pupil-linked arousal in mediating cognitive states, with pupil dynamics potentially serving as an indicator of transition between neural oscillatory patterns resulting from neuromodulation.

## 5. Future Neuromodulation Technology

The functional outcomes of understanding the unique role each neuromodulatory system plays in mediating cross-frequency coupling to facilitate certain brain functions have immense potential. Already, researchers have been able to develop neural interfaces to control and modulate a variety of neurological pathways [164]. More modern techniques that pay attention to multiple modalities of data are now employing LFP or EEG recordings in conjunction with neural stimulation technologies, including vagus nerve stimulation (VNS), transcranial direct current stimulation, and ultrasound stimulation [165,166,167,168,169,170,171]. Techniques such as focused ultrasound stimulation have the potential of delivering localized brain stimulation, allowing for non-invasive selective activation of smaller brain regions such as the LC, BF, or VTA [169,172]. Likewise, more invasive techniques such as CNS microsimulation with carefully designed patterns have enabled preferential activation of axons and somas, permitting a targeted approach to the activation of local neuronal circuits [173,174,175,176]. Additional invasive approaches are also currently in development, with clinical trials of optogenetics in humans being explored as a method to treat pain through selective neuronal activation, as well as through the activation of retinal ganglion cells [177]. While invasive methods will always provide the greatest specificity in neuronal activation, other non-invasive stimulation approaches are also proving to be effective. Both tDCS and TMS have been seen as appropriate treatments for depression and addiction, modulating prefrontal regions linked to cognitive symptoms [178,179]. Recently, studies have been utilizing current steering in conjunction with tDCS, enabling selective activation of deeper brain structures without interfering with superficial cortical regions [180,181].

While these methods are promising, VNS is proving to be one of the most effective approaches to non-invasive neuromodulation. The vagus nerve projects to a variety of brain regions, including neuromodulatory centers such as the LC, and VNS has been shown to be involved in phase–amplitude coupling [18,182]. These questions open the door to potentially developing the therapeutic paradigm in which VNS is conducted, using real-time neural oscillatory feedback as a form of optimization [18]. Additionally, other noninvasive stimulation methods, including transcutaneous vagus nerve stimulation, were shown to mimic VNS by modulating alpha EEG activity, suggesting that EEG can provide higher real-time feedback regarding arousal and cognitive changes [19,183]. Future adaptations of stimulation devices may need to incorporate time-variant phase–amplitude coupling, which could inform and decipher the role CFC plays in mediating or reflecting nervous system function [21]. While these enhancements may seem quite positive, one study showcased how theta–gamma cross frequency coupling induced via transcranial altering current stimulation actually worsened the ability of humans to employ cognitive control in goal-directed behavior [184,185,186]. It is essential for future devices, which aim to augment brain functions through the control of cognitive states, to optimize stimulation patterns based on real-time neural oscillatory dynamics to enable more informed treatments and feedback.

Future work reliably assessing real time cross-frequency coupling relationships in EEG and LFP signals could provide insight into how stimulation techniques can properly modulate cognitive function and behavioral outcomes. Experiments looking at key EEG features such as phase, amplitude, power, and frequency should be conducted in order to further explain how neural oscillations change in response to different cognitive processes. Selective neuromodulation of the noradrenergic, cholinergic, and dopaminergic system may provide insight into how different CFC phenomena occur, guiding our understanding into how complex synchronous neural networks communicate and formulate complex circuits to direct cognitive functions [187]. Closed-loop stimulation of these centers may provide a solution to uncovering the causality of CFC throughout the brain [9]. Neurological disease research for Parkinson’s, Alzheimer’s disease, schizophrenia, ADHD, and epilepsy may benefit greatly from this research, as neuromodulation could be used to change neural oscillatory patterns and network synchrony, potentially restoring or reversing erratic neural behavior caused by adverse neural synchronization [188].

## 6. Conclusions

The role of the neuromodulatory systems in influencing neural oscillations is an understudied topic. Decades of research have helped bring to light the relationship between neural oscillations and higher cognitive functions, but the causality behind these signals remains poorly understood. By reviewing the literature of each of the three major neuromodulatory systems (noradrenergic, cholinergic, and dopaminergic) and their representative cortical projections, effects on neural oscillations, and involvement in cross-frequency coupling, the complex nature of these neural computations can hopefully be further explained. To enhance our understanding of this relationship, it is essential to look towards specific oscillatory features such as cross-frequency coupling to understand the intricacies within large-scale neural synchronization. Studies emphasizing the importance of the activation and inhibition of these neuromodulatory centers in conjunction with frequency domain analysis could prove critical to understanding the causality of different behaviors and neurological disorders. These results could help inform the next generation of closed-loop neural stimulation devices with effective real-time outcomes, prioritizing the role of neuromodulation in controlling large-scale neural oscillations to restore or enhance brain functions.

## Figures and Tables

**Figure 1 biology-12-00371-f001:**
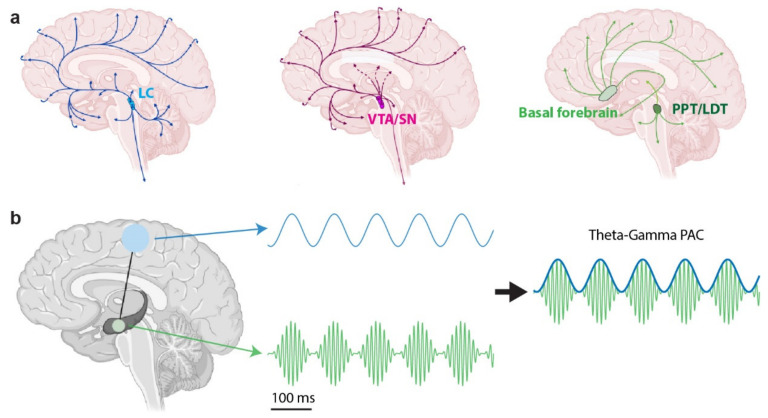
Anatomical representations of each neuromodulatory system and CFC visualization: (**a**) Anatomical locations and projections of the three neuromodulatory systems: the noradrenergic system (**left**), dopaminergic system (**middle**), and cholinergic system (**right**). (**b**) A cartoon illustrating theta–gamma phase–amplitude coupling (PAC).

**Figure 2 biology-12-00371-f002:**
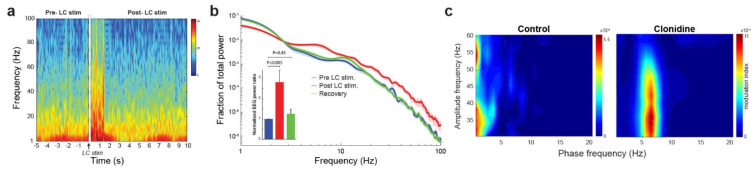
The causal effect of the noradrenergic system on neural oscillations: (**a**) Spectrogram of cortical EEG around phasic LC stimulation. (**b**) LC stimulation resulted in a significant increase in EEG power ratio in a high frequency (10–100 Hz) to low frequency (1–10 Hz); blue (pre LC stim), red (post LC stim), green (recovery); adopted from [105]. (**c**) Manipulation of the noradrenergic system through Clonidine administration, an alpha-2 agonist, altered phase-amplitude coupling of cortical EEG (2 mice, 9 sessions; unpublished data from the authors).

**Figure 3 biology-12-00371-f003:**
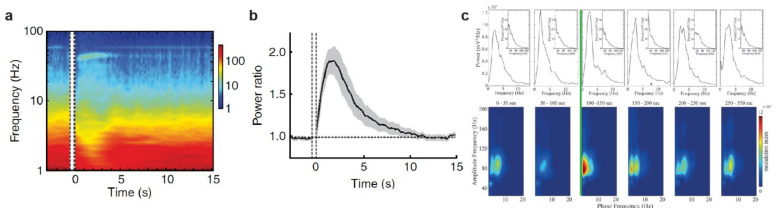
The causal effect of the cholinergic and dopaminergic system on neural oscillations: (**a**) Spectrogram of cortical LFP around nucleus basalis stimulation. (**b**) LFP power ratio (power at 10–100 Hz over power at 1–10 Hz) after nucleus basalis stimulation. Adopted from [128] with permission. (**c**) Dopaminergic receptor activation changed phase–amplitude coupling in mPFC. Adopted from [72].

## Data Availability

The data will be available from the corresponding author upon reasonable request.

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
