# Peer review of "Neuromodulation of Neural Oscillations in Health and Disease"

_biology, 2023, doi:10.3390/biology12030371_

Round 1

Reviewer 1 Report

The authors conducted a review on the impact neuromodulatory systems on neural oscillations with respect to cognition and neurologic diseases. The review is well written and organized. I have the following additional comments:

2.1. Frequency Spectral Analysis

The description of the frequency bands with respect to cognitive functions is well written but I think context is important for accurate interpretation. Can the authors specify the clinical scenario in which these frequency bands are associated with certain functions/states. For example, brain regions with epileptic foci or EEG recordings after neurosurgical intervention that disrupt the skull's integrity can also show increased power in these bands but would not necessarily be associated with the cognitive changes as described. The reason I ask for clarification is because the authors transition back and forth between discussing healthy state and pathological state but there are nuances to the characterization of neural oscillations. 

2.2. Cross-frequency Coupling

CFCs are described as both playing a pivotal role in normal cognitive function but also seen in pathologic states. Could the authors provide their interpretation of how the field should approach CFCs? Are they a surrogate biomarker of healthy neuronal function? But also pathologic function? Or do they represent communication in a non specific way?

3.2. The Cholinergic System 

Line 305 - The Aston-Jones study describes NE administration into the LC not the BF. Is this an error?

5. Future Neuromodulation Technology

Existing BCI-based neuromodulation in humans does not have the selectivity of brain stimulation compared to the lab setting. Can the authors comment on if this is necessary for future CFC analyses? For example, will we need something like optogenetics for humans to know that we are selectively modulation certain pathways that inform CFC analyses? This is especially relevant in non invasive stim such as tACS/tDCS where the stim field is so large/broad. 

Author Response

We sincerely thank the reviewers for their constructive comments and support. We believe that we have addressed all the concerns raised by the reviewers and strengthened the paper as a result. We have marked new text and text with modification blue in the revised manuscript to assist evaluation.

Below we provide a point-by-point response to their comments, with the reviewers’ original comments in black, and our responses in blue:

Reviewer 1:

The authors conducted a review on the impact neuromodulatory systems on neural oscillations with respect to cognition and neurologic diseases. The review is well written and organized. I have the following additional comments:

Thank you for your support.

2.1. Frequency Spectral Analysis

The description of the frequency bands with respect to cognitive functions is well written but I think context is important for accurate interpretation. Can the authors specify the clinical scenario in which these frequency bands are associated with certain functions/states. For example, brain regions with epileptic foci or EEG recordings after neurosurgical intervention that disrupt the skull's integrity can also show increased power in these bands but would not necessarily be associated with the cognitive changes as described. The reason I ask for clarification is because the authors transition back and forth between discussing healthy state and pathological state but there are nuances to the characterization of neural oscillations. 

Thank you for your suggestion. We have now included previous studies linking frequency bands to certain brain functions in clinical scenarios. We have also segregated previous work in healthy state and pathological state for clarity.

2.2. Cross-frequency Coupling

CFCs are described as both playing a pivotal role in normal cognitive function but also seen in pathologic states. Could the authors provide their interpretation of how the field should approach CFCs? Are they a surrogate biomarker of healthy neuronal function? But also pathologic function? Or do they represent communication in a non specific way?

Thank you for your comments. We completely agree with you that the interpretation of CFCs is a challenge as its causality is hard to test. We tend to believe that CFC is a predictive biomarker of communications between distinct populations of neurons mediating information processing, and is therefore subject to the influence of neuromodulatory systems. We indeed have some ongoing projects to test this hypothesis. We have included this speculation in the revised manuscript.

3.2. The Cholinergic System 

Line 305 - The Aston-Jones study describes NE administration into the LC not the BF. Is this an error?

Thank you for pointing this out. It was an error and has been fixed.

  1. Future Neuromodulation Technology

Existing BCI-based neuromodulation in humans does not have the selectivity of brain stimulation compared to the lab setting. Can the authors comment on if this is necessary for future CFC analyses? For example, will we need something like optogenetics for humans to know that we are selectively modulation certain pathways that inform CFC analyses? This is especially relevant in non invasive stim such as tACS/tDCS where the stim field is so large/broad. 

Thank you for your comments. It is important to note that optogenetics has been used in several clinical trials. Although the long-term safety of expressing ChR2 in the human nervous system, either through viral vectors or non-viral meaning, remains to be tested, the specificity provided by optogenetics technology will be unprecedented. However, in addition to optogenetics technology, carefully designed stimulation patterns may also be able to selectively stimulate the nervous system. For example, modeling results and experimental evidence demonstrated that biphasic current pulses with certain waveforms preferentially activate the soma of neurons than axons. There are some studies using the current steering approach to achieve selective activation using non-invasive stimulation such as tDCS/tACS. In addition, focused ultrasound neural stimulation may be able to provide non-invasive neural stimulation with reasonable selectivity. We have included this new information in the revised manuscript.

Reviewer 2 Report

Dear authors,

This is a very extensive review trying to address correlation of neural oscillations, cognitive function and role of different neurotransmitters systems within this frame. Although nicely written and very complete, it critically lacks structure and it is too long to read.

for example 2.2 CFC :

-      adapt this into separate subchapters that consider different cognitive functions: such memory/learning/ attention/task related?

-      Separate healthy conditions with pathological ones

the title states neuromodulation of neural oscillations, but the effects of neurostimulation on CFC and neurotransmitters is buried in the text. I would try to specifically address this separately and highlight the actual neurostimulation systems used in clinic, instead of referring to mostly experimental work.

The knowledge of tonic and phasic patterns of neurotransmitter release is an interesting point. Do you think different types of stimulation paradigms may affect these systems differently?

Add an anatomical figure that summarizes the different NT systems, potential correlating eeg coupling and role in cognition/other.

Author Response

We sincerely thank the reviewers for their constructive comments and support. We believe that we have addressed all the concerns raised by the reviewers and strengthened the paper as a result. We have marked new text and text with modification blue in the revised manuscript to assist evaluation.

Below we provide a point-by-point response to their comments, with the reviewers’ original comments in black, and our responses in blue:

This is a very extensive review trying to address correlation of neural oscillations, cognitive function and role of different neurotransmitters systems within this frame. Although nicely written and very complete, it critically lacks structure and it is too long to read.

Thank you for your support and constructive feedback. We have re-structured the manuscript to segregate discussions related to healthy and pathological conditions.

for example 2.2 CFC :

-     adapt this into separate subchapters that consider different cognitive functions: such memory/learning/ attention/task related?

-     Separate healthy conditions with pathological ones

Thank you for your suggestion. We have re-structured the manuscript to segregate discussions related to healthy and pathological conditions. In both healthy and pathological condition sections, we discussed CFC analysis in different cognitive functions. We believe that the change has significantly improved the clarity of the manuscript.

the title states neuromodulation of neural oscillations, but the effects of neurostimulation on CFC and neurotransmitters is buried in the text. I would try to specifically address this separately and highlight the actual neurostimulation systems used in clinic, instead of referring to mostly experimental work.

Thank you for your comment. Unfortunately, CFC based neuromodulation is still an under-studied field, and we are not aware of any FDA-approved CFC-based neurostimulator.  Although it has attracted much attention recently, the science linking neuromodulation, neural oscillations, and CFC remains to be explored, making this review heavily reliant on experimental work. This is actually the motivation for us to write this review paper. 

The knowledge of tonic and phasic patterns of neurotransmitter release is an interesting point. Do you think different types of stimulation paradigms may affect these systems differently?

Thank you for raising this interesting point. We do believe that tonic and phasic activation of the neuromodulatory systems have different functional consequences in brain functions. Many labs, including our lab, are working on this in attempt to delineate the functional difference between tonic and phasic release of neurotransmitters.

Add an anatomical figure that summarizes the different NT systems, potential correlating eeg coupling and role in cognition/other.

Thank you. We have added an anatomical figure of the three NT systems as well as a cartoon illustrating phase-amplitude coupling (Figure 1).

Reviewer 3 Report

Thank you for this nice review and overview,

if i could suggest one thing: i would mention a bit more about neuromodalation by devices like TMS, TDCS etc,

Author Response

We sincerely thank the reviewers for their constructive comments and support. We believe that we have addressed all the concerns raised by the reviewers and strengthened the paper as a result. We have marked new text and text with modification blue in the revised manuscript to assist evaluation.

Below we provide a point-by-point response to their comments, with the reviewers’ original comments in black, and our responses in blue:

Thank you for this nice review and overview,

 if i could suggest one thing: i would mention a bit more about neuromodalation by devices like TMS, TDCS etc,

Thank you for your support. We have added more clarifications of TMS and tDCS as well as studies of using them to treat clinical conditions such as depression and addiction.

Round 2

Reviewer 1 Report

The authors have adequately addressed my concerns/comments

Reviewer 2 Report

Thank yoou for your answers and revisions